# Patchwork of contrasting medication cultures across the USA

Rachel D. Melamed[1,2] & Andrey Rzhetsky [1,2,3]

Health in the United States is markedly heterogeneous, with large disparities in disease incidence, treatment choices and health spending. Drug prescription is one major component of health care—reflecting the accuracy of diagnosis, the adherence to evidence-based guidelines, susceptibility to drug marketing and regulatory factors. Using medical claims data covering nearly half of the USA population, we have developed and validated a framework to compare prescription rates of 600 popular drugs in 2334 counties. Our approach uncovers geographically separated sub-Americas, where patients receive treatment for different diseases, and where physicians choose different drugs for the same disease. The geographical variation suggests influences of racial composition, state-level health care laws and wealth. Some regions consistently prefer more expensive drugs, even when they have not been proven more efficacious than cheaper alternatives. Our study underlines the benefit of aggregating massive information on medical practice into a summarized and actionable form.

---

[1] Institute of Genomics, Genetics, and Systems Biology, Biological Sciences Division, Chicago 60637 IL, USA. [2] Section of Computational Biomedicine and Data-Intensive Science, Biological Sciences Division, Chicago 60637 IL, USA. [3] Department of Human Genetics, and Computation Institute University of Chicago, 900 E 57 St, KBCD 10160A, Chicago, IL 60637, USA. Correspondence and requests for materials should be addressed to A.R. (email: arzhetsky@uchicago.edu)

United States (US) society is famously and proudly multi-cultural and inhomogeneous. It can be viewed as a collection of almost disjoint communities that read distinct books and newspapers, watch different news channels, purchase distinct food items, and, when buying the same food ingredient, cook it in different ways. Thus, this patchwork pattern might be expected to extend to health care, specifically to prescription medications. Geographical disparities in health metrics across US regions are an active area of research[1,2]. Studies by the Institute for Health Metrics and Evaluation[3,4] show that inequalities in life expectancy across US counties are growing, and they are strongly influenced by county variation in socioeconomic, behavioral and health care factors. Using national survey and census data, they also find significant geographic variation in physical fitness[5] and disease burden[6]. Chetty[7] showed that the relationship between income and life expectancy was correlated with regional differences in population makeup and government spending. Murray[8] used geography and demographics to divide the country into eight Americas, and studied variation in health care, life expectancy and causes of death among the groupings.

Clinical information is increasingly captured in both electronic health care systems and in administrative claims databases. The coming wealth of observational health care data worldwide has motivated efforts to aggregate this data and use it to infer factors influencing health[9–12]. Here, we utilize observational data from over 150 million individuals in over 2000 counties across the United States to develop a nation-wide model to predict drug prescription. By analyzing how use of each drug departs from predicted values in each county, we obtain a measure of drug prescription comparable across all counties and drugs. We identify meaningful variation in use of drugs, which we show is associated with demographic and geographic differences between counties. These contrasts delineate sub-Americas characterized by different disease collections and, for patients with the same disease, distinct medications. Previous studies have mainly used national surveys information to infer health disparities: this is the first study to repurpose coded health care data to uncover geographical variation in medical care. Despite using only prescriptions, and not geographic or socioeconomic information, we recover known regional variation. But the comprehensive, data-driven nature of our design reveals previously unreported similarities and differences between regions of the country and highlights major sources of variation in prescribing preferences.

## Results

### A model for drug prescription, and deviation from the model.
In this study, we chose to focus on predicting which medications are prescribed to a particular patient, rather than prescription duration or dosage of medications. We refer in the following to first-time drug prescription, meaning a unique first incidence of prescription in a person's record. First-time drug prescription rates are related to both burden of disease and to the clinical choices of care providers. Although total prescription rates of a drug may also be of interest, these are heavily influenced by the amount of care a patient receives.

We examine prescription of drugs in a subset of the Truven MarketScan claims data, containing millions of individuals each followed for up to 10 years (from 2003 to 2013). The present analysis focuses on prescriptions for female patients who have county codes and multiple years of drug prescribing information. Prescription records include drugs dispensed, prescription week and age at prescription. From this data, we formulate a base model quantifying the probability that a patient will have a first-time prescription for a given drug, accounting for age, calendar year and amount of medical attention (Fig. 1a). We fit the base model separately for each drug, using data from all counties, containing a total of around 36 million patient-years. Using a held-out set of patients, we confirm that this model can generate unbiased predictions of drug use (Supplementary Figure 11).

Using the nation-wide base model, we calculate the expected incident prescription for the 598 most highly prescribed drugs in 2334 populous counties. We compare the expected number of prescriptions against the number observed for each drug in each county, quantifying the difference in a value we call the drug-county deviance. A high positive deviance for a given drug in a given county indicates more prescription than expected, given the nation-wide data on use of that drug, and the distribution of medical records in that county. For example, if a drug were only used in one county, that county would have a high positive deviance for that drug, and every other county would have a negative deviance for the drug. The drug-county deviance represents un-modeled sources of variation between the counties.

### Drug-county deviance is consistent with known variation.
To assess whether the deviance measure represents meaningful signal, we compare drug-county deviance with other sources of information. We show that drug use is significantly more similar for pairs of counties located closer together (Spearman's $\rho = 0.30$, $p < 10^{-30}$, p-value of Spearman coefficient is calculated as described in Methods), and pairs of counties with similar demographics (Spearman's $\rho = 0.31$, $p < 10^{-30}$, demographic data described in Methods). State-level influences such as insurance networks and state legislation would be expected to impact drug use, and we also are able to detect this effect ($p < 10^{-18}$, regression coefficient $F$-test, described in Methods). Next, we create state drug deviance values, which measure the disparity between observed and expected prescriptions for each drug in a state. Clustering states by the similarity of their profiles of drug use recovers known similarities between states (Fig. 1b). We also find some unexpected groupings—for example, northern New England (Massachusetts, New Hampshire, Vermont, and Maine) is most similar to Minnesota and Wisconsin. The correlations suggest that drug use particularly differs between northern and southern states, and between urban and rural states.

Much as similar counties have similar drug preferences, we find that drugs from the same Red Book therapeutic class have more similar use across counties ($p < 10^{-3}$, described in Methods, Supplementary Figure 1a). For some drug classes, such as thyroid hormone replacement, drugs have extremely similar deviance across counties (Supplementary Fig. 1b).

Other classes contain drugs with differing characteristics, and we detect curious diversity in drug use. The class of opioid analgesics includes some Drug Enforcement Agency schedule II compounds, with highest potential for abuse among legal drugs. Among these, oxycodone deviance ranges widely across regions, as has been reported[13] (Supplementary Figure 2). Curtis et al. showed that variation in schedule II opioid prescription, in the mid-2000s, was related to state prescription drug monitoring programs, intended to deter abuse[14]. Our results are consistent with these effects (rank-sum test $p < 0.008$, described in Methods, Supplementary Fig. 2). Other contemporary studies[15–17] suggested that schedule II monitoring programs resulted, undesirably, in increased utilization of schedule III drugs. Consistent with these reports, we find negative correlation of schedule II drug deviances with schedule III hydrocodone and propoxyphene opioids.

We also compare disease-associated drug use with disease-associated death rates[18]. Use of drugs is consistent with the varying burden of these diseases across regions (Supplementary Fig. 3b, Supplementary Table 1). For example, the counties with

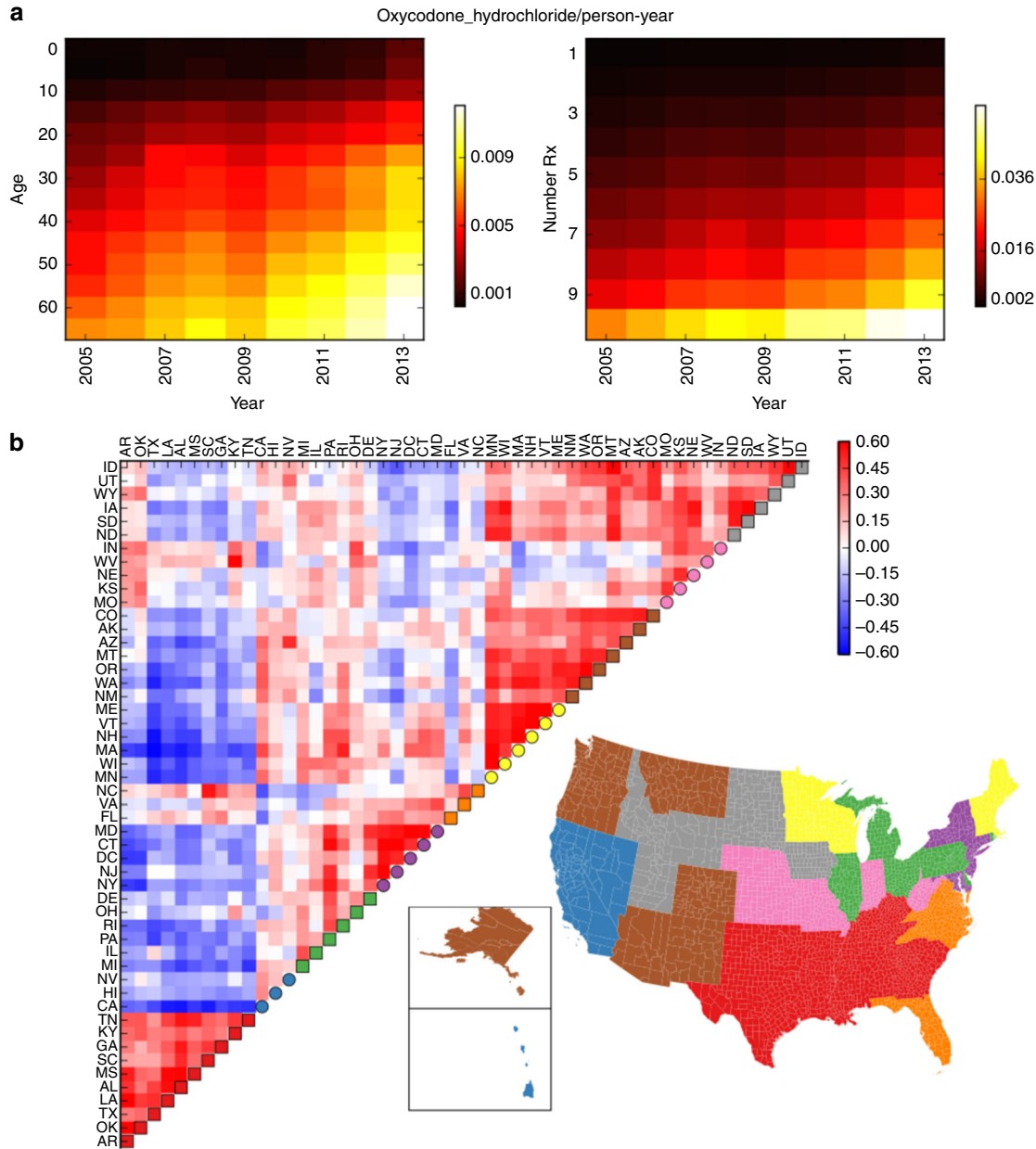

**Fig. 1** Quantifying variation in drug use, and comparing states by departure from expected drug use. **a** The probability that a person who could be prescribed oxycodone hydrochloride will have a record for this prescription, is shown as a function of year, age and number of prescriptions in that person-year. This probability increases over years. **b** We generate drug deviance vectors for each state, which represent how much more or less of each drug is used in the state as compared with the nation-wide model. Clustering states by their Spearman correlation in these vectors (see color scale) recovers known similarities between states. We create non-overlapping clusters (legend for each state appears on diagonal) and color the map of the country by drug cluster

the highest use of antidiabetics have the highest diabetes death rate.

Both the analyses of opioids and of death rates reveal meaningful covariation of drug deviance that is consistent with other regional differences. The analysis of state similarities (Fig. 1b) shows that areas with similar populations have similar profiles of drug use, suggesting that the United States contains multiple subcultures of drug prescription. Thus, in the next section we aim to identify the chief types of variation in drug use across counties.

**Covariation in drug use across counties**. Correlations between use of medications arise due to their complex relationships with

diseases and treatment choices. For example, obesity rates vary between counties, and obesity leads to multiple secondary health problems indicating use of certain drugs. An intuitive approach to untangle the signal in this data is to find the axes of strongest covariation underlying these correlations. These represent continuous latent variables, and we would expect them to relate to factors such as obesity that explain the most variation in drug use. We perform principal component analysis (PCA) of the matrix of drug-county deviances. PCA finds as its first component a projection (a linear combination) of the county drug data that captures the most variance among counties. The second component is another linear combination of the drug values that maximizes the variance, with the constraint that it is uncorrelated with the

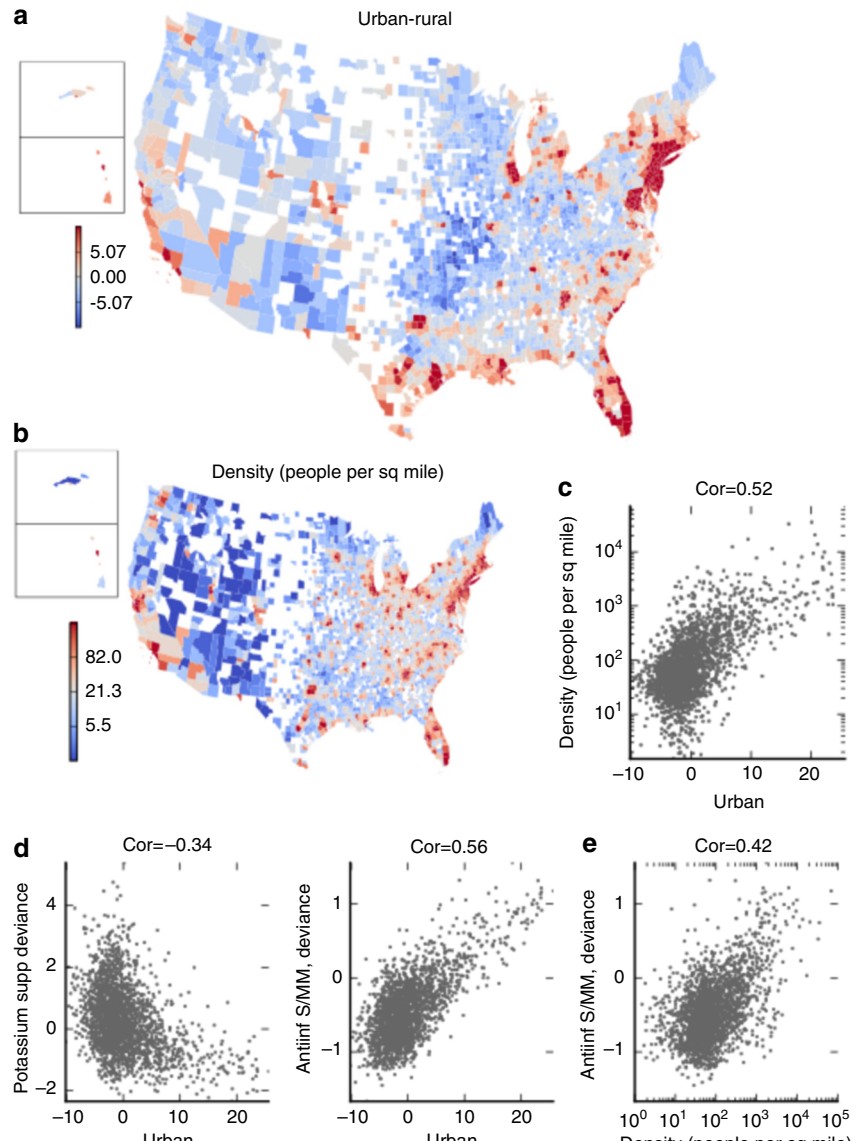

**Fig. 2** Visualizing variance captured by Urban–Rural. **a** We project each county's drug deviance vector onto the component, and map the projected values. The high values have a clear correspondence with urban centers in the USA. **b** Map of US population density per county. Colors are shown in a log scale (see legend). **c** Each point is a county, and the projected Urban–Rural value is compared with the population density in that county, with a Spearman correlation (abbreviated, cor) of 0.52. **d** Each county's drug deviance value for a class of drugs is compared with the Urban projection value. A positive drug deviance represents more drug used in a county than expected. An example of a negatively correlated class is potassium supplements, typically used for people on potassium-depleting thiazide diuretics. The class Antiinf S/MM consists of dermatological topicals for skin inflammations, such as acne. **e** Directly compares the demographic characteristic from **c** with the drug class from **d**

first, and so on. We project each county onto the most significant components recovered; these represent a few orthogonal rankings, or scores, of counties. We confirm that PCA results are robust by assessing the effect of removing subsets of the data (Supplementary Fig. 4).

Using these projections as proxies for independent latent variables influencing drug use, we examine the implications of the strongest four PCA components, which jointly explain 77% of the variance. We give each projection a nickname summarizing the positively and negatively correlated county characteristics. The first component (35% of variance), nicknamed North/West-Southeast, varies most between counties located in Southeastern states as opposed to Northern and Western states; the second component represents the Urban–Rural axis (20% of variance); the third, South/West-Northeast, divides out mainly northeastern

counties with high white population, high obesity, high fraction of population insured and high health costs (12% of variance); the fourth, White/Wealth-nonWhite/Poverty particularly captures drug use associated with income, and fraction of non-Hispanic African American, versus non-Hispanic White population (10% of variance). Thus, the top four components reflect multiple factors that vary across counties, and that independently influence prescription.

We visualize the variation captured by each PCA axis in two ways. First, we display how the projected value of each county on this component varies with the county's geographical location, using a colored map of the country (Figs. 2a, 3a, Supplementary Figs. 6, 7). Second, we compare the projection to demographic information on each county (described in Methods, Supplementary Fig. 5). Figure 2a–c show the association of Urban–Rural

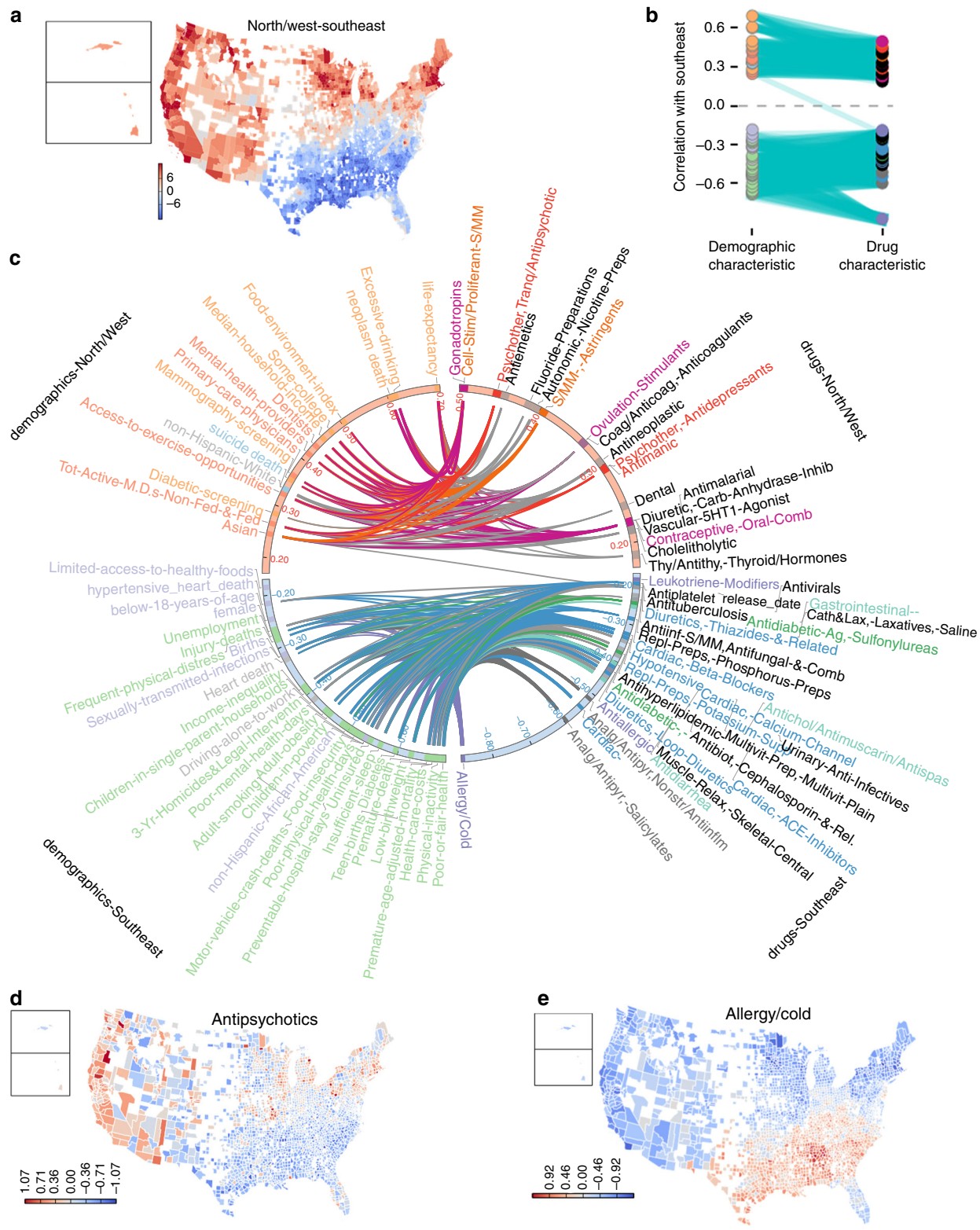

with population density. We also compare Urban–Rural projection against the county's averaged deviance value for drugs in various therapeutic classes. Figure 2d shows that as Urbanity grows, counties have fewer prescriptions for potassium repletion supplements, corresponding to lower use of potassium-depleting diuretics prescribed for hypertension. Urban counties use more

dermatologic agents, such therapeutic class Anti-inflammatory Skin/Mucous Membrane (Antiinf S/MM), as well as fertility medications.

Another latent variable, North/West-Southeast, reflects a pattern of covariation among counties that appears geographically distinct from that of Urban–Rural (Fig. 3a). In order to

**Fig. 3** Visualization of North/West-Southeast. **a** Map of county projected values. **b** This plot continues the idea of the visualization in Fig. 2 by showing the correlations of county values for the projected dimension, with demographic indicators, and the drug class deviances. On the left side, the correlations between North/West-Southeast and demographic factors are shown. Correlated demographic variables that co-cluster in Supplementary Fig. 5 have the same color. On the right side, drug therapeutic classes that correlate positively or negatively are shown. These are colored by related therapeutic uses (see **c**). Cyan lines connect demographic characteristics that are positively correlated with drug characteristics, with a correlation cutoff of > 0.2. **c** This plots the same bipartite graph as in part **b**, but in circular format to allow each demographic or drug class to be labeled. Thus, the closer a node is to the top or bottom pole, the stronger the positive or negative correlation with North/West-Southeast. Edges are colored according to the drug class they connect to. **d** Use of antipsychotics, showing lower deviance values in the southeastern counties. **e** Antipsychotics are negatively correlated with allergy and cold medicine

summarize this component, we compare positively and negatively correlated demographic characteristics and drug therapeutic classes, arranged by correlation with North/West-Southeast (Fig. 3b, c). Counties in the Southeast extreme have poorer health and higher non-Hispanic African-American populations, as compared with North/West counties. All seven drugs belonging to the therapeutic class of antipsychotics are prescribed more in North/West counties (mean $\rho = 0.46$, Fig. 3d). Antipsychotic prescription is correlated with county life expectancy ($\rho = 0.31$) and excessive drinking ($\rho = 0.34$), among other demographic characteristics that are also associated with North/West counties. It is unsurprising that more wealthy and healthy counties could have increased diagnosis and thus treatment of psychotic disorders, as compared with poorer parts of the country with other health problems. In contrast, allergy and cold medicines (Fig. 3e) are highly used in most Southeast counties. This is consistent with a pattern of less prescription of preventive care drugs. People in these counties likely have a greater proportion visits for acute problems. Southeast counties are also distinguished by demographic characteristics including obesity and diabetes, explaining their greater use of drugs for obesity-related illnesses including hyperlipidemia, hypertension and diabetes.

Figure 3b, c show that the demographic characteristics most positively correlated with the North/West-Southeast axis, and the drug classes most positively correlated with this component, are also positively correlated with each other. This indicates that the differences between counties in terms of drug use are consistent with demographic differences. This is notable because no demographic data were used in the PCA. The consistency between drug-county deviance and demographic indicators is supported by a canonical correlation analysis, which finds a number of significant canonical correlates between the drug-county deviance matrix and the county demographics matrix (described in Methods, Supplementary Table 2).

Like the first two components, the next two also have a strong geographical association. For South/West-Northeast (mapped in Supplementary Fig. 6a), the fraction of population insured is higher in the Northeast than South/West. Some preventive care therapeutic drug classes are prescribed more in the Northeast, and these are generally prescribed more in counties with higher insurance rates. This includes fertility medications, fluoride treatments and smoking cessation drugs such as varenicline (Supplementary Fig. 6b, c). This axis of variation captures some state-level covariation in drug use. In Supplementary Figure 6a and d, counties in Massachusetts, Vermont and Minnesota have a higher projected value for South/West-Northeast, compared with counties across the state line in neighboring states. Notably, only these three states had current laws mandating public disclosure of payments from pharmaceutical companies to prescribers, which could have some shared influence on prescribing patterns[19–21].

For the fourth dimension, White/Wealth-nonWhite/Poverty, the map of projections reveals that counties on the nonWhite/

Poverty extreme are particularly located in the poor rural South, in a strip from the Carolinas west through Mississippi (Supplementary Fig. 7). At the White/Wealth end, counties are located in the middle of United States, and they tend to spend more on health care. They have particularly high prescription of antidiabetic and antihypertensive drugs, consistent with high rates of obesity and diabetes in the nonWhite/Poverty counties (Supplementary Fig. 7e). Another characteristic is a lower rate of prescription for the three thyroid hormones. Deviance values for thyroid hormones and diabetes drugs are negatively correlated (Spearman's $\rho = -0.19$, $p = 6.3 \times 10^{-21}$). Hypothyroidism can also co-occur with, and exacerbate, obesity and diabetes[22–25], making under-diagnosis of hypothyroidism a possible concern.

To illustrate the geographical associations, we assign each county to at most one of the top four components, if that county has an extremely high or low projected value on that component (Fig. 4). This visualization is somewhat cartoonish, as counties often have extreme projections on more than one component— for example, Marin County, California scores high on North/West, Urban and South/West. Bronx County, New York, projects as Urban, Northeast and nonWhite/Poverty.

**Geographical variation in use of expensive branded drugs**. Because there are wide regional differences in drug spending in the United States[26], we investigate variation in drug use associated with generic versus brand-only availability. From Medicaid's database, we obtained drug generic status and approximate price (National Average Drug Acquisition Cost (NADAC))[27]. In Fig. 5a, we show variation in use of anti-infective and anti-inflammatory eye drops. From left to right, the drugs are more positively correlated with Urban–Rural, and are more expensive: the correlation between Urban–Rural association and brand-only availability is 0.89. The most expensive anti-inflammatory eye drop is around double the price of the second-most expensive. We contrast the most urban against the most rural counties (selected counties shown in Fig. 5c), to visualize their use of generic versus branded drugs in different therapeutic classes (Fig. 5d).

Of course, other factors besides price drive drug selection. But, across drug classes ranging from antihypertensives to skin care, Urban counties prescribe more brand-only drugs (Figs. 5b, c, Spearman's $\rho = 0.48$, $p = 8.7 \times 10^{-37}$, described in Methods). We summarize correlations between price preference and each of the top four variables, for the largest drug therapeutic classes. Figure 5b shows that Urban–Rural, North/West-Southeast and White/Wealth-nonWhite/Poverty have consistent price preferences, though the trend is apparent in different drug classes for each axis. Northeast counties also somewhat prefer brand drugs, but less consistently across all drug classes.

Like Urban counties, Southeast counties tend to prefer expensive drugs (Spearman's $\rho = -0.095$, $p = 0.019$), even accounting for drug class (Fig. 5b). In addition to their overall lower use of antidepressants and antipsychotics, Southeast counties particularly prescribe less of cheaper psychiatric drugs

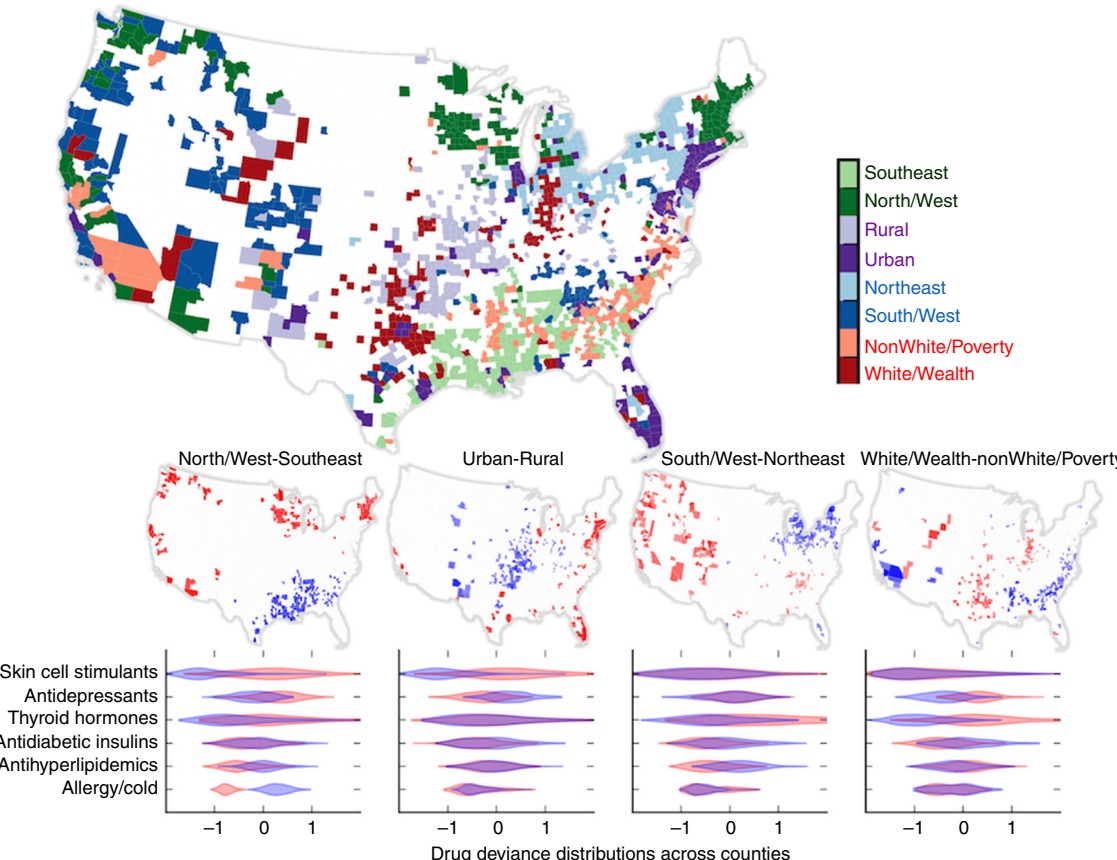

**Fig. 4** Summary of the first four components. Each county is assigned to one of the components if it falls into the top or bottom tenth percentile of a component. The map on top shows these assignments, with one hue per component, and lighter color indicating counties with extreme negative projections, darker for extreme positive. The maps in the middle show the same extreme counties, separated into each component. The bottom panel shows the distribution of deviances for some chosen therapeutic drug classes. The overlapping violin plots depict the use of the drugs in the class for counties in the high (red), and low (blue), extremes of each component

(Fig. 5e). The preference for more expensive drugs is surprising, as Southeastern counties are generally poorer than the national average. But of these, the poorest counties have a low projection on White/Wealth-nonWhite/Poverty. Opposed to the general trend for Southeast counties, nonWhite/Poverty projection is correlated with preference for generic drugs (Spearman's $\rho = -0.23$, $p = 1.4 \times 10^{-8}$).

**Variation in price preferences versus evidence of effectiveness.** Prilosec was a blockbuster drug for heartburn that went off-patent in 2001. That year, Prilosec's maker introduced Nexium, a much more expensive brand-only drug. Nexium has a different form of the same active ingredient, and any superiority to Prilosec remains controversial[28]. As would be expected of competing drugs for the same condition, use of Prilosec, and its generics, is negatively correlated with use of Nexium across counties (Spearman's $\rho = -0.22$, $p = 9.9 \times 10^{-28}$). In Fig. 5f, we contrast use of these two drugs across counties ranked by the three components most correlated with price.

As another example, we examine variation in prescriptions for hypertension drugs, which comprise 10% of the drugs in our analysis. These represent a wide range of treatment choices. Near the beginning of our study period, revised guidelines recommended cheaper older drugs, thiazide diuretics, as more beneficial than expensive new hypertension treatments[29–31]. However, later studies showed that physicians resisted re-adoption of older thiazide diuretics, possibly due to marketing of newer drugs[32]. One estimate[33] suggests this resulted in $1.2 billion per year of unsupported excess prescription spending. We find that cheaper drugs within classes (Supplementary Fig. 8), and cheaper classes of antihypertensive agents (Supplementary Fig. 9), are prescribed more in regions preferring generic drugs. Notably, Urban counties use more angiotensin II receptor blockers, the most expensive antihypertensive class. In contrast, nonWhite/Poverty counties have high hypertension drug use across many classes, but they particularly use cheaper drugs, such as non-combination thiazide diuretics.

**Modeling regional price preferences.** As a different approach to examining brand preferences in each county, we model drug deviance as a function of drug generic status and therapeutic class. This model estimates preference for generic or brand drugs in each county (Fig. 6a), and state (Fig. 6b). The result is consistent with our previous findings: urban areas, particularly the corridor from New York to Washington, DC prefer more expensive drugs, as do parts of the southeast. Northern New England, some Midwestern and western states prefer cheaper drugs. Using the demographic information on each county to predict preference for expensive drugs, we find income, health care costs and access to exercise opportunities are most predictive of expensive drug preference (Fig. 6c). However, this model only explains a quarter of the observed variance in brand preference across counties ($R^2 = 0.24$). Other factors, such as state-level laws, insurance networks or cultural preferences, likely explain the rest. The latent variables uncovered via PCA point to possible influences.

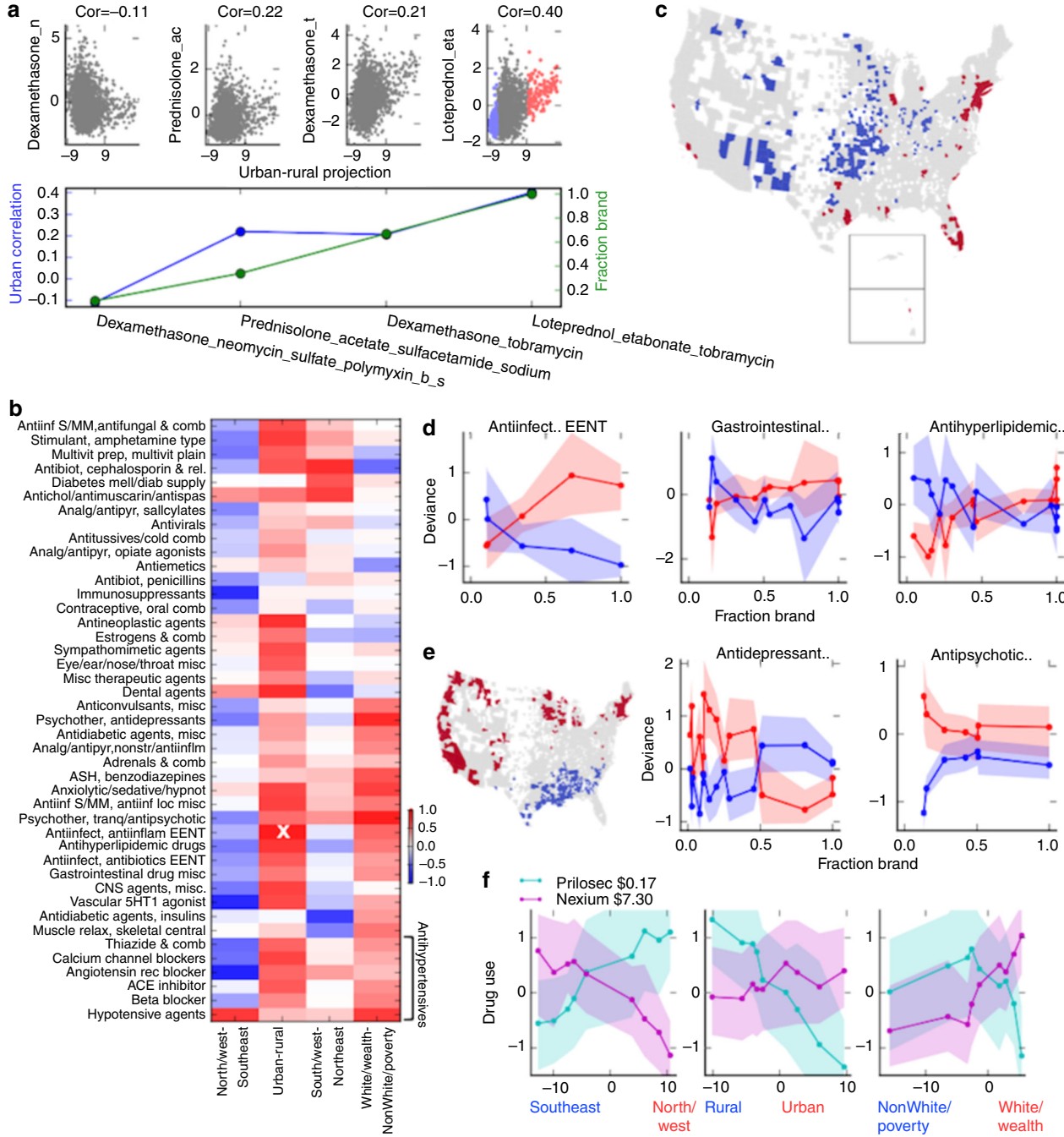

**Fig. 5** Correlation of drug price with regional drug preferences. **a** In the top panel, anti-inflammatory eye drops (part of class Antiinfect, Antiinflam EENT (ears, eyes, nose throat)), are arranged from left to right by fraction generic. Each scatter plot compares county Urban–Rural projection against deviance in the given drug. The correlation (abbreviated, cor) increases with fraction generic: shown for the same drugs in the bottom panel. **b** The correlation between price and drug preference, for the largest therapeutic classes. The example from part (**a**) is highlighted with a white X. **c** Urban and rural counties, also highlighted in the rightmost plot in (**a**). The two extreme county groups are contrasted in part (**d**). **d** In each panel, the points are drugs in a therapeutic class. The drugs are arranged by increasing fraction brand-only (x axis). Fraction brand is compared with drug deviance values in the Urban (red) and Rural (blue) counties (see **c**). Filled regions highlight the 25–75 quartiles for use of each drug across each set of counties. The left plot shows the class from (**a**). **e** Analogous to **c**/**d** but for North/West-Southeast, and different drug classes. **f** Each panel compares use of Prilosec and Nexium across counties projected onto one component. Counties are binned by the component projection. The x axis denotes the component projection, and the y axis shows use of the drugs in those counties. The filled regions show 25–75 percentile of the deviance values across counties in the bins

## Discussion

We have developed and supported a systematic approach to compare use of all popular prescription medications across most counties. Our approach enables emergent detection of multiple types of variation between counties, reflected in the components of the PCA. Particularly intriguing is evidence that some regions appear to prefer more expensive branded drugs. Health costs in the United States account for >17% of the nation's economy, and drugs are the fastest-growing category of health care spending[34]. Open questions include whether preference for generics reflects more efficient, but equally effective, use of health care dollars, and if pharmaceutical marketing is swaying treatment choices.

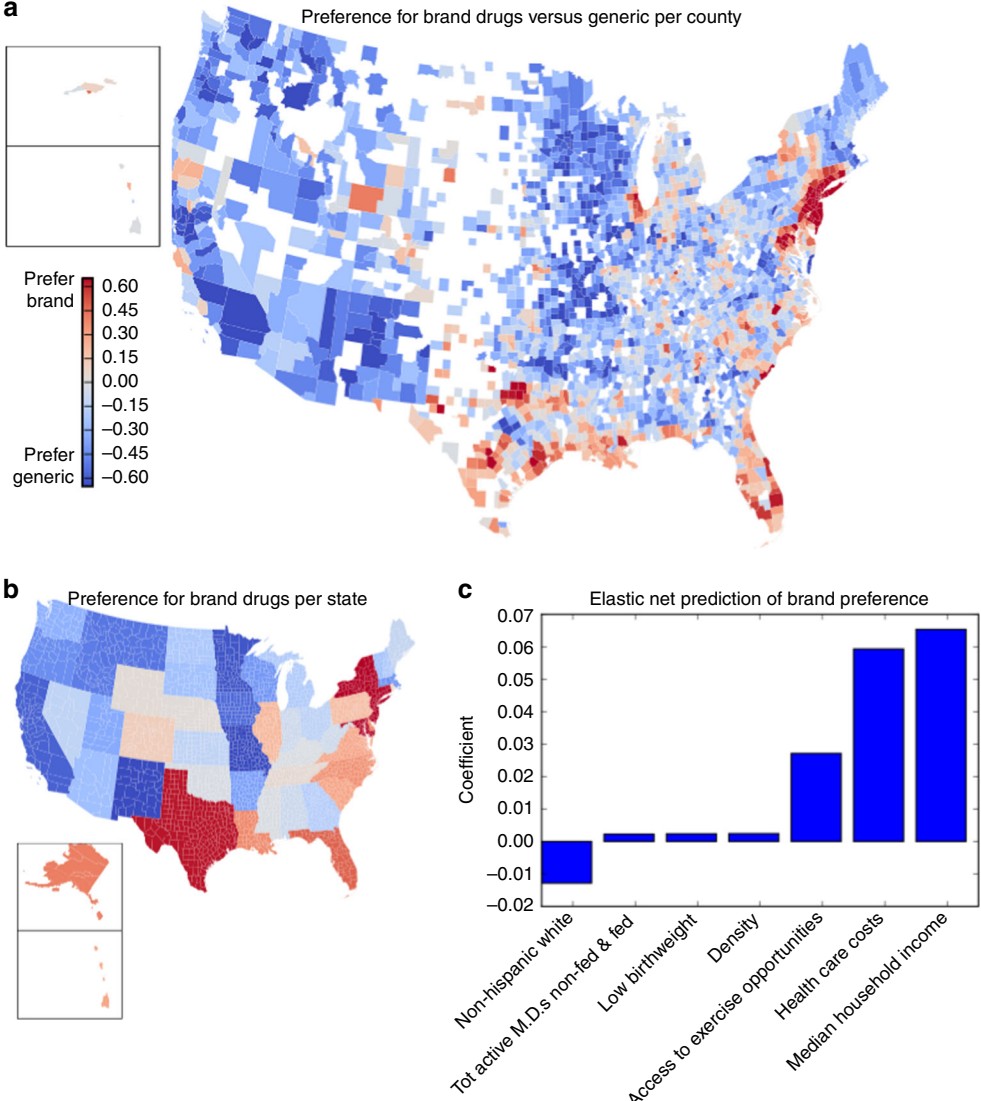

**Fig. 6** Estimation of brand preference per county and state. **a** We plot the regression coefficients for brand preference per county. Positive coefficients imply a preference for brand drugs, and against generic drugs. **b** The same, for states. **c** Comparison of county brand preference against demographic characteristics, using an elastic net regression model to predict brand preference

Our data do not uniformly sample across different areas of the country, and it do not uniformly represent sociodemographic strata within each county. Truven's MarketScan data have been previously used to investigate incidence of disease[35,36], and geographical variation in diagnosis[37–39]. The data have also been used to survey prescription practices[40,41]. Comparison of prescription rates from Marketscan data against health records data have shown that the two data sources are comparable[11]. Although missing prescriptions in claims data have been documented[42], these are greater in elderly populations, and they are often associated with patient records that are entirely missing (rather than missing single prescriptions)[43]. Our work carefully conditions all results on the total number of drugs prescribed to each patient. We pose our question therefore in the following way: given private insurance, and given a level of utilization of this insurance, how does drug use vary? This yields a generalizable model applicable to other health care data. The axes of variation described in this work are similar when the analysis is repeated on males (Supplementary Fig. 13) and they are largely stable over multiple years (Supplementary Figure 14), despite changes in health care options, and drug availability, over this time span.

The canonical correlation analysis indicates a number of significant canonical variables beyond those we explored here (visualized at drugmap.uchicago.edu, preview available in Supplementary Fig. 15) inviting future work. Another way to build on our framework would incorporate diagnosis data to compare treatments in patients with a given medical condition across the country. We could also use drug-county deviances in slices of time to examine how medical practices spread through regions. As data grow, temporal changes, as well as age-related changes, in diagnosis and drug use will become increasingly informative. In addition to large-scale trends, our results (tables at https://figshare.com/projects/Patchwork_of_contrasting_medication_cultures_across_the_USA/36311 and code at https://github.com/RDMelamed/county-drug-variation) enable investigation of geographic variation in prescription of hundreds of specific drugs—we have chosen only a few for purposes of illustration. Researchers could use these estimates to find areas underserved with regard to a drug of interest, either to understand the causes of these disparities or to target regions for intervention. The impacts of some state-level policies can also be investigated, as in our analysis of opioids. Related work[44] has used IMS health prescription sales data to propose drug use as a proxy

for disease rates, enabling more geographically uniform surveillance of health needs, such as in rural areas. Quantifying the heterogeneity in health factors is essential for researchers who wish to estimate the effects of health interventions. To this purpose, the patchwork of counties we identify could provide a starting point for subdividing the diverse USA population into subgroups with relatively homogeneous risks of disease and medical care.

## Methods

**Predicting rate of drug use from claims data.** The Truven Health Analytics MarketScan data (recently acquired by IBM Watson) contains patient identifiers linked to time-stamped National Drug Codes (NDC), as well as information about patient age and zip code. The study was reviewed by Institutional Review Board of the University of Chicago and found exempt because it deals with existing de-identified data. Counts of total observations and observations per group of drugs are shown in Supplementary Table 3. We match NDC codes to drug generic names using the MarketScan RED BOOK™ Supplement (includes variables related to drug prescription). For each drug generic name, we separately model the probability of a new prescription (incident prescription) over the course of a person-year. We require that a person must have >2 years of data, where the first 2 years are used only for comparison to ensure that we are measuring incident drug use. Our sample unit is the person-year, where we assume that each observed person-year has a homogeneous probability of drug use that depends on four factors: age (divided into 5-year bins), calendar year, the number of new prescriptions for that person in that year and the number of years observed with any new prescription. Although number of prescriptions in a calendar year might indicate the current level of medical attention, number of years with any new prescription reflects the consistency of care over time. An example of the influence of this variable is shown in Supplementary Figure 10a.

Referring to one setting of these four variables as a bin denoted $b = \{age = a, year = y, number\ of\ new\ medications = r, number\ of\ new\ medication\ years = m\}$, we obtain all person-years falling into a given bin. We can then model the probability of use of drug $d$ for a given bin $b$ with the following assumption:

$$p(\text{take drug } d \text{ while in bin } b) = \frac{y_{d,b} : \text{individuals observed taking } d \text{ in } b}{n_{d,b} : \text{individuals observed in } b} \#. \quad (1)$$

This is equivalent to a discrete-time survival analysis[45], which is simply a logistic regression that models over multiple time points the probability that a person will have an event, given that a person is in the risk set (to use the survival analysis term) for the event. Here, the event is new prescription of $d$, and the risk set is comprised of subjects observed in bin $b$. A person is observed in the risk set in bin $b$ for drug $d$ (and thus counts toward the denominator $n_{d,b}$) if the following two conditions are met: (1) the person is observed in the year $y$, of age $a$, etc, and has $r > 0$ new prescriptions during that year; and (2) that person has never been observed taking drug $d$ before. As each bin is one combinatorial setting of these four variables, there are over 30,000 observed bins. Some bins involving patients with many prescriptions per year can be very sparsely observed; thus, we collapse bins for highly observed patients (Supplementary Fig. 10).

We fit this model using the four bin variables as categorical (dummy) covariates, and we allow all pairwise interactions between the variables. We use the sklearn toolkit to fit the model, with a regularization parameter tuned for best fit by cross validation. We call this model the USA model. In order to assess the bias of this model, we use a set of holdout samples and we compare predicted drug use in each sample against the actual number of each drug prescribed. We conclude that our model is unbiased for prediction of drug use in a population (Supplementary Fig. 11 shows some examples).

**Calculation of drug deviance value per county.** Next, we compare the model to the data from each county to determine if each county has more or less drug prescribed than would be expected in the USA model. Let $n_{d,b,c}$ be the number of people observed in county $c$, who could have taken drug $d$ in bin $b$. Of these people, we observe $y_{d,b,c}$ who actually took that drug. The model already accounts for variation in age and amount of medical care, but in order to ensure that different population distributions in different counties do not influence the results, we also standardize all populations to the nation-wide population, as follows. We calculate a weight value for each bin, representing the fraction of the national population that falls in that bin: $w_{d,b} = \frac{n_{d,b}}{\sum_{b'} n_{d,b'}}$. Then, for each drug, and for each county, we obtain the following weighted values for the observed number of people total, and taking the drug, respectively: $n_{d,c} = \sum_b w_{d,b} \cdot n_{d,b,c}$ and $y_{d,c} = \sum_b w_{d,b} \cdot y_{d,b,c}$. We compare this with the number of prescriptions expected under our model. For one bin, before weighting, we would predict $n_{d,c,b} \cdot \widehat{p_{d,b}}$ new prescriptions, where $\widehat{p_{d,b}}$ is the predicted probability from the logistic regression model for drug $d$, in the bin $b$ (age, calendar year, etc). Thus, for county $c$, the corresponding population-standardized expected value of drugs taken under the USA model is $\hat{y}_{d,c} = \sum_b w_{d,b} \cdot n_{d,c,b} \cdot \hat{P}_{d,b}$.

Now, we compare the observed and expected county values: $\hat{y}_{d,c}$ and $y_{d,c}$. Although the ratio of these values provides a decent estimate of the relative use of

the drugs, the ratio is very sensitive to sampling variability for less common drugs. Therefore, we instead measure the departure of observed drug use in a county from the expected drug prescription using the deviance residual of the county value from the binomial model of nation-wide drug use [46]. This measure quantifies how well the USA model fits the county data by evaluating the likelihood of the county data under the USA model. Then, drug-county deviance for drug $d$ in county $c$ is:

$$G_{d,c} = \text{sign}\left(y_{d,c} - \hat{y}_{d,c}\right) \cdot \left[ y_{d,c} \cdot \log\frac{y_{d,c}}{\hat{y}_{d,c}} + \left(n_{d,c} - y_{d,c}\right) \cdot \log\frac{n_{d,c} - y_{d,c}}{n_{d,c} - \hat{y}_{d,c}} \right]^{\frac{1}{2}} \#. \quad (2)$$

Compared with the log-ratio, the deviance residual has much less extreme values for less common drugs (Supplementary Fig. 12).

**Aggregating other county characteristics.** In order to compare drug deviance values with other measures that vary across counties, we compile data on health and demographic characteristics of counties from the County Health Rankings report[47]. These demographic county-level indicators include racial composition, medical care availability, income, health status and behavioral measures. To this, we add population and density data, longitude, and latitude from the US Census[48,49], life expectancy estimates from the Institute on Health Metrics and Evaluation[3,5], and death rate data from the CDC[18]. We download both the age-adjusted total death rate per county and the age-adjusted death rate per county attributed to each of the top causes of death. We divide these to obtain each county's fraction of deaths due to a cause. The full list of demographic characteristics and sources is shown in Supplementary Table 4. We remove a number of characteristics with >10 missing values, but we keep the 69 demographic variables with fewer than 10 missing values across the 2334 most populous counties. For these, we impute the demographic value by a regression using other counties as observations, and all other demographic variables as predictors.

**Drug characteristics.** In addition to mapping NDC to generic name, Red Book also contains other information for each drug. This includes therapeutic class, an indicator of whether this is an over the counter product versus prescription-only, if the primary use is acute or chronic, the Drug Enforcement Agency classification, and indicator of whether NDC corresponds to a generic or a brand medication. We collapse each generic name drug into a summary of these characteristics. We also obtain from Medicaid the NADAC[27]. Finally, we download drug information from the Food and Drug Administration[50], with each drug's release date and pharmacological class. Generic status, release date and NADAC are, as expected, very correlated with each other.

**Comparison of drug-county deviance with known variation in counties.** As a first point of positive control, we compare counties with each other in terms of our estimate of their drug use. We define a drug deviance vector for each county, containing a profile of the positive or negative deviance from the expected value of each drug for that county. We calculate the Euclidean distance between each pair of counties in terms of standardized drug deviances. To show that this distance reflects known differences between counties, we compare pairwise distances with the other county covariates. First, we compare the pairwise drug distances with the geographical distance between the county pairs, calculated using the Euclidean distance between their latitude and longitude. We also compare the drug distance scores with distance between their normalized vectors of demographic information, described above. We assess whether the drug distance is correlated to these independent measures of distance between counties using Spearman correlation. For each correlation, we also report in the Results the $p$-value from the t-transformation of the correlation, as implemented in scipy[51].

To assess whether we capture the effect of state-level factors, we examine pairs of neighboring counties, where at least one of the counties falls on a state border. First we identify neighboring counties, using USA county shape data (http://bokeh.pydata.org/en/latest/_modules/bokeh/sampledata.html), which contains the counties as polygons. We identify bordering county pairs as those for which the line segments intersect. We match counties to their state, and to Census and demographic information, using county Federal Information Processing Standards (FIPS) codes.

We model the pairwise distance between the two drug deviance vectors as a linear function of the county pair's demographic distance and a binary indicator of whether the pair of counties fall in the same state. This data fit well to a model with statistically significant coefficients for both variables, in the expected directions. That is, a pair of bordering counties with less similar demographics (greater distance between their demographics vectors) have greater distance between their drug deviance vectors (demographic distance coefficient = 0.16, standard error = −0.06, $p = 0.01$). If the two counties are in the same state, they have a smaller distance between their drug use vectors (same-state indicator coefficient = −1.95, standard error = 0.22, $p < 10^{-18}$). These results point to a meaningful correspondence between drug deviance characteristics of a county and other known characteristics of counties.

**Comparing county deviance vectors for similar drugs**. Next, we evaluate whether a drug's vector of deviances across counties captures meaningful information about that drug. Using the same matrix of drug-county deviance, we obtain a county deviance vector for each drug. We compare classification of drugs by Red Book therapeutic class against an unsupervised hierarchical clustering of the drug deviance vectors across counties, using the Adjusted Rand index[52]. The Rand index compares a known classification (in this case, the therapeutic classes) to a learned classification (in this case, the clusters based on county deviance). The score evaluates how well a clustering retains in the same cluster each pair of items belonging to the same class. We use the Adjusted Rand index, implemented in sklearn adjusted_rand_score, which corrects for the number of co-clustered items expected by chance. There are 153 therapeutic classes, and some of these are highly overlapping: that is, many drugs fall into both classes. We collapse these 153 classes into 75 classes with minimal overlap. Only 46 of our 598 drugs are in more than one collapsed class. Rand index is not quite ideal for our purpose for two reasons: it is designed to evaluate recovery of non-overlapping classes, and it is not traditionally used for such a large number of classes.

To adapt the Adjusted Rand index, we create 100 versions of the therapeutic class assignments, where we randomly assign each of these drugs to only one of the therapeutic classes it belongs to. We evaluate the distribution of Adjusted Rand index values across these versions of the known class assignments (Supplementary Fig. 1). Traditionally, an Adjusted Rand index of 1 indicates good recovery of true classes, but this is not realistic with so many classes, and in addition our purpose is not to classify drugs but to evaluate the signal in our drug deviance vectors. In order to evaluate whether the resulting Adjusted Rand index values represent a significant co-clustering of drugs in the same class, we compare the scores with randomly permuted class assignments, over 1000 permutations. We find a strong separation between the Rand index between the true and random class assignments (Supplementary Fig. 1). This shows that drugs that have the most similar prescription trends across counties in fact have similar therapeutic purpose.

**Analysis of variation in opioids prescription**. We compare the state-level drug deviance for opioids with the findings of Curtis et al.: they found that AK, AZ, DE, MD, MA, NH, SC and TN had the highest rates of schedule II oxycodone uses, and CA, TX, IL, MI and NY had lower use of these drugs. We performed a rank-sum test for each group of states to assess whether our results agree with theirs. We find that the low-use states show significantly lower deviance values (rank-sum test, $p = 0.002$) and the high-prescription states have higher deviance values ($p = 0.008$) (Supplementary Fig. 2d).

**Regularized regression analyses**. To predict the thyroid drugs, using all other drugs, we fit an elastic net regression model using the python sklearn package[53]. We use Z-scored values of all other drugs as predictors so we can directly compare the regression coefficients. The regularization shrinks most of the regression coefficients to zero, and for each of the three regressions, the remaining thyroid drugs are the most predictive drugs.

We take a similar approach to examine the consistency between drug prescription and fraction of deaths in a county due to each cause of death. We again use elastic net regression, with cause of death per county as the outcome, and all drug-county deviance values as candidate predictors. Again, we obtain resulting regression coefficients that can be compared across drugs. Next, we assess whether drugs that treat a cause of death are particularly predictive of that cause. We curate sets of drugs for chronic care that are obviously related to causes of death (Supplementary Table 1). For example, although there are drugs to treat flu and pneumonia, this set of drugs is not generally for chronic care and is not as specific to these conditions as, for example, antihypertensive or antidiabetic drugs. Again, most regression coefficients are shrunk to a narrow distribution around zero. For each drug set, we assess if the coefficients in the set are significantly high or low using a two-tailed rank-sum test (Supplementary Table 1, Supplementary Fig. 3). For each cause of death with an obviously related set of drugs, we find that the related drug set has the most positive set of coefficients.

To predict use of thyroid drugs as a function of demographic indicators per county, we use sklearn's MultiTaskElasticNet with thyroid hormone deviance as the outcome. We standardize the demographic variables, and we report the variables that depart from zero in Supplementary Figure 7.

**Dimensionality reduction**. We perform the PCA using singular value decomposition with sklearn[53], after normalizing the data (centering and variance standardizing the drugs). We use the resulting eigenvectors to transform the county drug deviance data into the projected space. Additionally, we perform canonical correlation analysis on the drug-county deviance data and the county demographic data. This approach treats each county as an observation on these two sets of variables. The canonical correlation analysis identifies linear combinations of the variables in the two data sets (drug data and demographic data) that maximize the cross-covariance between the two sets. As canonical correlation analysis is only recommended when the sample size is many times larger than the number of variables, we restrict this to the 197 drugs with more than half a million users across the country, and we use only the fully observed demographic features, filtering highly collinear features. Using the R package CCA[54], we obtain the canonical

correlates, and we test significance using another R package, CCP[55]. The results (Supplementary Table 2) suggest more than a dozen significant canonical correlates.

To assess the stability of the PCA components, we select subsets of the data. We compare the eigenvectors found when using the full matrix of all drugs with the vectors for the submatrix with a therapeutic class of drugs removed. Comparing each of the top dimensions by the dot product of the eigenvectors, we observe how often the dot product is near 1 (Supplementary Figure 4). For the first two dimensions, the dot product ranges between 0.99 and 1 for all of the 153 classes removed. For second two, 147 of 153 drug classes have eigenvector dot products >0.95. The class of drugs with the highest influence are the analgesic opioids.

**Code availability**. All code to make the figures and perform statistics cited in the text is available at https://github.com/RDMelamed/county-drug-variation. Figures containing maps of the United States were drawn using the python package basemap and shape files from the United States Census Bureau, http://www2.census.gov/geo/tiger/GENZ2010/gz_2010_us_050_00_5m.zip.

## Data availability

The summarized data sets derived from the MarketScan data and used to make the figures in this article are available at https://figshare.com/projects/Patchwork_of_contrasting_medication_cultures_across_the_USA/36311. In particular, this contains the matrix of drug-county deviance values that were used to perform the main analyses.

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

## Acknowledgements

This work was funded by the DARPA Big Mechanism program under ARO contract W911NF1410333, by National Institutes of Health grants 5K01ES028055, R01HL122712, 1P50MH094267 and U01HL108634-01, and by a gift from Liz and Kent Dauten.

## Author contributions

R.D.M. and A.R. designed experiments, R.D.M. analyzed data, and R.D.M. and A.R. wrote the manuscript.

## Additional information

**Competing interests:** The authors declare no competing interests.

