## [Peer Review File · Nature Communications]

Reviewers' comments:

Reviewer #1 (Remarks to the Author):

This paper applies sophisticated statistical analysis techniques to an impressively large data set, to study geographical variation in drug utilization across the United States. This is an interesting and novel investigation.

In general, I found the manuscript well-written and interesting. The presented results are well-supported by the methods and the data, and the figures provide insight into the conclusions.

I do have a couple of concerns.

First, the authors mention "prediction" in several places, and on L293 there is a claim of "a generalizable model". It seems that the model is built on the full data set and hence it is not clear that there is truly a "predictive" model being evaluated, but rather a model that is fit to the data. L335 refers to a "random test sample" used for tuning, but it is not clear that this test sample is independent of the data used to build the model, nor is it clear that this tuning set is independent of the various samples described on L336-342. The authors conclude on L347 that their model is unbiased. I would be more convinced if the data sets that are considered were more clearly independent. The authors could simply refer to modeling, rather than prediction; in that case the claims about predictiveness and lack of bias might need to be reconsidered.

Second, there are a number of additional points that should be clarified in the manuscript, as listed below:

- Why was the study restricted to females?
- L106-108: The logic here should be clarified. I suppose more death by diabetes means less death by other causes. Why is suicide specifically relevant here?
- L469: I assume that the category of "thyroid drugs" was derived from the RedBook classes but this should be stated explicitly.
- L199: Is there evidence for "high rates of obesity and diabetes" in these counties that can be cited? (In any case, I would suggest moving the phrase "consistent with ... diabetes" to the end of the sentence)
- L218: How are drug prices obtained? (Is this from NADAC, mentioned on L397? This could be stated more directly.)
- L290: Be more explicit about limitations on sampling
- L295: the link to <http://52.44.9.226/app> did not work for me; can some be included in supplementary material?
- L312: I believe this should be "the number of `_new_` prescriptions ..." / L314: similarly, I believe r is defined as new medications in year y . Please define these more carefully, including what "new" means exactly.
- L314: What is the intuition for splitting people with differing numbers of (new) prescriptions and differing numbers of observation years into different bins? Would it make more sense to organize people in terms of total numbers of prescriptions?
- L329: what is the threshold for collapsing "highly observed" patients? (generally, insight into the distribution here, as alluded to in previous comment, might be helpful)
- L315: some background on survival analysis (citations, details, some

explanation of how the authors' approach is similar or different) is warranted.

- L384: Is there a threshold for missing values?
- L405: What kind of distance metric is assumed?
- L438: Is there a citation/motivation for using the Rand Index? (The authors may be interested to also consider the Adjusted Rand Index, Adjusted Mutual Information and the more recent Standardized Mutual Information)
- L469: I assume that the set of thyroid drugs is determined by Red Book class?

Typos/Edits:

L32 : inhomogeneous -> heterogeneous

L84: northern New England is not a state; be more specific.

L113: identify chief -> identify the chief

L128: the recovered most significant components -> the most significant components recovered

L276: This is conjecture. Please use "may explain" or "likely explains"

L503: the double negative here is rather confusing ...

Reviewer #3 (Remarks to the Author):

In order to make claims about identification of 'medication cultures' within the United States during recent years, the authors provide a fascinating manuscript that integrates data from multiple sources, using what appears to be a novel data analysis approach. The approach and findings are novel and should be of interest to the research community served by Nature Communications.

There is important related work that has not apparently been considered. Here, I draw attention to published work based on the IMS Health data systems on pharmaceutical products used in office-based practices, pharmacies, hospitals, and health plans, as well as other proprietary sources. It is possible that the authors have drawn data elements from these sources, but as noted below, there is some ambiguity in the description of the research approach.

It is difficult to judge the impact this work might have on the field. There is far too little information provided about the data sources and the research approach, including details of probability and statistics that often are quite specifically tied to data sources. As a practicing researcher, I would not be able to reproduce what they have done based on the specifications given in this manuscript, and I do not know any independent research team that would be able to do so.

In summary, I would judge the manuscript as a work in progress, but the progress made to date is not sufficiently transparent for this reviewer to have confidence that these results would be reproduced by others if they had access to the same sets of data, even if the research approach were described with more clarity than presently is the case.

Reviewer #4 (Remarks to the Author):

The manuscript describes a very thorough approach to analyze patterns of drug consumption in the US. The authors can only present a small sample of the results that can be generated, without

attempting a detailed analysis of one drug or one group of drugs, which would be more interesting from a clinical and an economic point of view.

The main limitation of the manuscript is in my opinion the lack of a discussion of the assumptions and choices made in building the resource (e.g., ignoring multiple prescriptions per year for a give drug for a given patient) and the limitations in the available data (e.g., effect of missing data).

The number of subjects included in the analysis and the number of prescriptions, including that of major drug groups (e.g., in a supplementary table) should be indicated.

Since the data cover a 10-yr period, it would be interested to assess whether some of the main results are stable over time or trends can be identified.

Reviewer #5 (Remarks to the Author):

This study entitled Patchwork of contrasting medication cultures across the USA seeks to examine different geographical patterns in prescribing in US based on medical claims data. The paper is based on a very large dataset from over 150 million individuals in over 2,000 counties across the US and the overall aim is to develop a nationwide model to predict drug prescription.

Unfortunately, the paper fails to deliver on this very ambitious aim.

The paper is structured in a rather unusual way. Thus, the results and discussion sections come before the methods. However, the authors do mix methods and results within the result section and sometimes even start the discussion within the result section. The overall discussion is very brief and inadequate. The authors, however, highlight one of the key issues with their data that it does not uniformly sample across different areas of the country, and it does not uniformly represent sociodemographic strata within each county. The results of their analyses are therefore questionable.

The paper includes several sub-studies, but none of these have been sufficient described or discussed. As it is this paper is not suitable for publication.

Reviewer #1 (Remarks to the Author):

This paper applies sophisticated statistical analysis techniques to an impressively large data set, to study geographical variation in drug utilization across the United States. This is an interesting and novel investigation.

In general, I found the manuscript well-written and interesting. The presented results are well-supported by the methods and the data, and the figures provide insight into the conclusions.

I do have a couple of concerns.

First, the authors mention "prediction" in several places, and on L293 there is a claim of "a generalizable model". It seems that the model is built on the full data set and hence it is not clear that there is truly a "predictive" model being evaluated, but rather a model that is fit to the data. L335 refers to a "random test sample" used for tuning, but it is not clear that this test sample is independent of the data used to build the model, nor is it clear that this tuning set is independent of the various samples described on L336-342. The authors conclude on L347 that their model is unbiased. I would be more convinced if the data sets that are considered were more clearly independent. The authors could simply refer to modeling, rather than prediction; in that case the claims about predictiveness and lack of bias might need to be reconsidered.

1. Since we repeated our analysis for males for point 2 (below), we have made sure to hold out 250000 men from the creation of the model. After the model was fit, we used it to predict drug prescription for these 250000 male patients: we partitioned them into 5 groups and compared the predicted and observed use of a number of drugs for each group. This was a much simpler evaluation (and more relevant) than our previous evaluation, which used the model to predict prescriptions in a randomly chosen year. The results (Supplementary Figure 11) show that for a random sample of patients across the country, who were not used to build the model, we are able to make an unbiased prediction of incident drug prescription.

Second, there are a number of additional points that should be clarified in the manuscript, as listed below:

- Why was the study restricted to females?

2. Because females generally have a differing set of health needs from males, we did our initial work in females. However, we expected that geographic sources of variation would be very similar for males. In order to address this comment, we repeated the main analysis, and the PCA for males. A sketch of the results, which are very similar to the results for females, is shown in the new Supplementary Figure 13. We will include the results for males in the GitHub repository as a resource for researchers interested in the differences between the sexes.

- L106-108: The logic here should be clarified. I suppose more death by diabetes means less death by other causes. Why is suicide specifically relevant here?

3. We have rewritten that paragraph in order to make it clear that the goal of the analysis is to show that use of drugs is related to external data on burden of disease.

- L469: I assume that the category of "thyroid drugs" was derived from the RedBook classes but this should be stated explicitly.

4. We clarified that the therapeutic class indeed came from RedBook.

- L199: Is there evidence for "high rates of obesity and diabetes" in these counties that can be cited? (In any case, I would suggest moving the phrase "consistent with ... diabetes" to the end of the sentence)

5. We have fixed the sentence and included in Supplementary Figure 7e a map showing rates of obesity in the united states, which are from the demographic data compiled as described in Methods.

- L218: How are drug prices obtained? (Is this from NADAC, mentioned on L397? This could be stated more directly.)

6. We have clarified in this line that drug prices and generic status come from NADAC.

- L290: *Be more explicit about limitations on sampling*

7. See also point 26. We have added a description of possible concerns regarding prescription sampling in the Discussion section, as well as added a Supplementary Table 3 showing the sample sizes.

- L295: *the link to .. did not work for me; can some be included in supplementary material?*

8. We have updated the link to drugmap.uchicago.edu, as well as included a screenshot in the supplementary materials (Supplementary Figure 15). The code itself to create the app is in the GitHub repository with the rest of the materials needed to reproduce the results (<https://github.com/RDMelamed/county-drug-variation/tree/master/app>).

- L312: *I believe this should be "the number of _new_ prescriptions ..."* / L314: *similarly, I believe r is defined as new medications in year y. Please define these more carefully, including what "new" means exactly.*

9. The reviewer is correct and we have fixed this sentence. We have rewritten this section to make it clearer what the predictor variables are and to define "new prescription" more clearly.

- L314: *What is the intuition for splitting people with differing numbers of (new) prescriptions and differing numbers of observation years into different bins? Would it make more sense to organize people in terms of total numbers of prescriptions?*

10. Generally, we expect them to measure different things. While total number of new prescriptions might represent how sick the person is in a short period, number of observation years shows that they are receiving regular care. To support this, we have included a new Supplementary Figure 10a showing an example of a drug that is more commonly prescribed among people with more years of records, even when number of prescriptions per year is fixed.

- L329: *what is the threshold for collapsing "highly observed" patients? (generally, insight into the distribution here, as alluded to in previous comment, might be helpful)*

11. We thank the reviewer for this suggestion, which we have included as a new Supplementary Figure 10b,c. We bin patients receiving more than 8 prescriptions per year into bins of [9-12, 13-18, and >19] prescriptions per year.

- L315: *some background on survival analysis (citations, details, some explanation of how the authors' approach is similar or different) is warranted.*

12. We have expanded on this in the text and included a reference to the classic paper on discrete time survival analysis.

- L384: *Is there a threshold for missing values?*

13. Fewer than 10 missing values. We have made this clear in the text.

- L405: *What kind of distance metric is assumed?*

14. We use Euclidean distance. This is now stated in the text.

- L438: *Is there a citation/motivation for using the Rand Index? (The authors may be interested to also consider the Adjusted Rand Index, Adjusted Mutual Information and the more recent Standardized Mutual Information)*

15. We have improved this section of the text by including a citation and describing it a bit more. We also clarified that we are actually using the Adjusted Rand index.

- L469: *I assume that the set of thyroid drugs is determined by Red Book class?*

16. Yes, see number 4.

Typos/Edits:

L32 : *inhomogeneous -> heterogeneous*

17. Fixed

L84: *northern New England is not a state; be more specific.*

18. Fixed

19. Fixed *L113: identify chief -> identify the chief*

20. Fixed *L128: the recovered most significant components -> the most significant components recovered*

21. Fixed *L276: This is conjecture. Please use "may explain" or "likely explains"*

22. Fixed *L503: the double negative here is rather confusing ...*

Reviewer #3 (Remarks to the Author):

In order to make claims about identification of 'medication cultures' within the United States during recent years, the authors provide a fascinating manuscript that integrates data from multiple sources, using what appears to be a novel data analysis approach. The approach and findings are novel and should be of interest to the research community served by Nature Communications.

There is important related work that has not apparently been considered. Here, I draw attention to published work based on the IMS Health data systems on pharmaceutical products used in office-based practices, pharmacies, hospitals, and health plans, as well as other proprietary sources. It is possible that the authors have drawn data elements from these sources, but as noted below, there is some ambiguity in the description of the research approach.

23. We agree that the goals of the IMS Health research is related to the area explored in this paper. We did not use any data from IMS health: their data appear to derive from pharmacies, while ours derive from insurance claims. However, we did cite a study which made use of this data (McDonald 2013). We have aimed in this revision to make it very clear which data sources were used in the analysis. In the discussion, we have added a reference (Cossman 2010) connecting our work to a related work from IMS health.

It is difficult to judge the impact this work might have on the field. There is far too little information provided about the data sources and the research approach, including details of probability and statistics that often are quite specifically tied to data sources. As a practicing researcher, I would not be able to reproduce what they have done based on the specifications given in this manuscript, and I do not know any independent research team that would be able to do so.

24. We agree that reproducibility is extremely important in all studies. In order to address this concern, we have aimed to clarify our approach to creating the statistical model in the Methods section. We have also included a new Supplemental Table 4 detailing each data source used to profile demographic characteristics of the counties. In addition, we have included in the Method a link to a GitHub repository that allows recreation of all statistics, figures, and tables in the paper (in particular, <https://github.com/RDMelamed/county-drug-variation/blob/master/figures.ipynb>). The table of demographic data is available in this repository. We believe that someone with access to the MarketScan data would be able to recreate our results, and a researcher without access to MarketScan could use our supplementary materials, including the GitHub code, to recreate most of the figures and other results in the text.

In summary, I would judge the manuscript as a work in progress, but the progress made to date is not sufficiently transparent for this reviewer to have confidence that these results would be reproduced by others if they had access to the same sets of data, even if the research approach were described with more clarity than presently is the case.

25. We are confident that our results are fully reproducible, as all source codes related to our analysis were made publicly available.

Reviewer #4 (Remarks to the Author):

The manuscript describes a very thorough approach to analyze patterns of drug consumption in the US. The authors can only present a small sample of the results that can be generated, without attempting a detailed analysis of one drug or one group of drugs, which would be more interesting from a clinical and an

economic point of view.

The main limitation of the manuscript is in my opinion the lack of a discussion of the assumptions and choices made in building the resource (e.g., ignoring multiple prescriptions per year for a given drug for a given patient) and the limitations in the available data (e.g., effect of missing data).

26. We have expanded on the reasoning for our choice of research topic in the first paragraph of the results section. We do not consider this to be an assumption but a choice of research topic. We have added discussion of the data source, including its limitations, in the Discussion section.

The number of subjects included in the analysis and the number of prescriptions, including that of major drug groups (e.g., in a supplementary table) should be indicated.

27. We thank the reviewer for this suggestion, which we have included as Supplementary Table 3. We have also mentioned the number person-years at the end of the second paragraph of Results. Note that although 36 million person-years forms the denominator of our rate prediction, more data than this is involved: for example, we require a two year period of observation with no history of the drug for each patient, in order to declare a drug to be a new prescription.

Since the data cover a 10-yr period, it would be interested to assess whether some of the main results are stable over time or trends can be identified.

28. We agree that a temporal analysis of trends in drug use would be an interesting avenue for future research. Although we have chosen to focus on geographic variation in drug use, in the Discussion we mention possible lines of inquiry related to temporal changes.

In order to address this point, we have added a new Supplementary Figure 14, which was created as follows. Using new drug prescriptions occurring only in each of five single-year periods, we calculate drug-county deviances, and the resulting principal component analysis. Our main findings seem to hold across the slices of time points. This is despite the smaller sample size involved in each time point, and despite changes in populations, health policy, and drug availability across years. The main differences are positive-negative direction of the components (the sign of the direction is not meaningful), and other rescalings of the importance of the components. There is other interesting variation over time. The biggest evolution over time occurs in the Northeast-South/West component, which is consistent with its apparent reflection of state-level effects. For example, the implementation of health care reform in Massachusetts in 2006 might account for some differences between the (2005, Northeast-South/west) plot and the (2007, Northeast-South/west) plot. Dissection of which drugs vary between these time points, in selected regions, is possible with our method, and as we mention in the Discussion, this is a possible avenue for future research.

Reviewer #5 (Remarks to the Author):

This study entitled Patchwork of contrasting medication cultures across the USA seeks to examine different geographical patterns in prescribing in US based on medical claims data. The paper is based on a very large dataset from over 150 million individuals in over 2,000 counties across the US and the overall aim is to develop a nationwide model to predict drug prescription. Unfortunately, the paper fails to deliver on this very ambitious aim.

The paper is structured in a rather unusual way. Thus, the results and discussion sections come before the methods. However, the authors do mix methods and results within the result section and sometimes even start the discussion within the result section. The overall discussion is very brief and inadequate. The authors, however, highlight one of the key issues with their data that it does not uniformly sample across different areas of the country, and it does not uniformly represent sociodemographic strata within each county. The results of their analyses are therefore questionable.

The paper includes several sub-studies, but none of these have been sufficient described or discussed.

29. The structure, clarity, and style of the manuscript were substantially improved to address the reviewer's comments. Importantly, we included results of model validation on a held-out sample. The results are very good, hopefully, enough to convince this reviewer.

We have also expanded our discussion of sampling limitations of our data set (see also point 7 and 26 above). Additionally, our revision has improved the description of our methods (see also point 24 above). We hope that our revision satisfies these concerns.

Reviewers' Comments:

Reviewer #1:

Remarks to the Author:

The authors have done a very good job of addressing my previous comments; I appreciate the additional experiments and analysis that they have done to support the validity of their model and to expand the analysis to males.

I commend the authors on a solid and interesting piece of work on such a large-scale data set.

Reviewer #4:

Remarks to the Author:

N/A

Reviewer #5:

Remarks to the Author:

I am still concerned about the validity of this study.